# Optimizing Neural Networks via Koopman Operator Theory

**Akshunna S. Dogra**[*]
John A. Paulson School of
Engineering and Applied Sciences
Harvard University
Cambridge, MA 02138,
asdpsn@gmail.com

**William T. Redman**[*]
Interdeparmental Graduate Program in
Dynamical Neuroscience
University of California, Santa Barbara
Santa Barbara, CA 93106
wredman@ucsb.edu

## Abstract

Koopman operator theory, a powerful framework for discovering the underlying dynamics of nonlinear dynamical systems, was recently shown to be intimately connected with neural network training. In this work, we take the first steps in making use of this connection. As Koopman operator theory is a linear theory, a successful implementation of it in evolving network weights and biases offers the promise of accelerated training, especially in the context of deep networks, where optimization is inherently a non-convex problem. We show that Koopman operator theoretic methods allow for accurate predictions of weights and biases of feedforward, fully connected deep networks over a non-trivial range of training time. During this window, we find that our approach is >10x faster than various gradient descent based methods (e.g. Adam, Adadelta, Adagrad), in line with our complexity analysis. We end by highlighting open questions in this exciting intersection between dynamical systems and neural network theory, and additional methods by which our results may be generalized.

## 1   Introduction

Despite their black box nature, the training of artificial neural networks (NNs) is a discrete dynamical system. During training, weights/biases evolve along a trajectory in an abstract weight/bias space, the path determined by the implemented optimization algorithm, the training data, and the architecture. This dynamical systems picture is familiar: many introductions to optimization algorithms, such as gradient descent (GD), visualize training as a process where weights/biases change iteratively under the influence of the loss landscape. Yet, while dynamical systems theory has provided insight into the behavior of many complex systems, its application to NNs has been limited.

Recent advances in Koopman operator theory (KOT) have made it a powerful tool in studying the underlying dynamics of nonlinear systems in a data-driven manner [1–8]. This begs the question, can KOT be used to learn and predict the dynamics present in NN training? If so, can such an approach, which we call *Koopman training*, afford us benefits that traditional NN training approaches, like backpropagation, cannot? This viewpoint was recently proposed in work by one of the authors, which demonstrated that KOT and NN training are intimately connected [9]. While significant effort has been dedicated to using NNs to discover important features of KOT [10–12], this work was, to our knowledge, the first application of KOT for the training of NNs (see Sec. 1.1).

An appealing aspect of KOT is that it is a linear theory for nonlinear dynamical systems. In our case, this means that evolving NN weights/biases using Koopman operators involves only matrix

---

[*]Authors contributed equally

computations. We show, under certain mild assumptions, how these computations can be considerably cheaper than standard NN training methods. Thus, we show how successful application of KOT to NNs could lead to significantly accelerated training techniques. We demonstrate this potential in the context of feedforward, fully connected NNs. We emphasize that our methods are applicable to a wide range of NNs beyond the specific ones we study - indeed they should generalize well to a broad variety of systems that make use of iterative optimization methods.

This work is structured as follows. We begin with a brief introduction to KOT. We describe and deepen the existing argument that NN training can be viewed as a process that evolves weights/biases through the action of a Koopman operator. We then discuss different implementations of Koopman training, introducing a novel approach of partitioning the Koopman operator approximation. This idea is crucial for ensuring that the run time complexity of Koopman training is comparable or better than that of the standard training methods. We then present the results of Koopman training two different feedforward, fully connected, deep NNs: an NN differential equation (DE) solver and a classifier trained on the MNIST data set. These NNs show significant variation in architecture sizes, objectives, optimizers, learning rate, etc, lending credence to our assertions that Koopman training is a versatile and powerful technique. Our basic Koopman training implementations successfully predict weight/bias evolutions over a non-trivial number of training steps, with computational costs one to two orders of magnitude smaller than the standard methods we compared to. We end by discussing future problems of interest in Koopman training and the potential more advanced KOT methods offer in extending our results.

## 1.1   Related work

Earlier this year, one of the authors showed that the renormalization group (RG), a widely used tool in theoretical physics, is a Koopman operator, and that KOT could help speedup computations of relevance in the field (e.g. critical exponent calculations) [13]. These results were achieved by viewing the RG as a discrete dynamical system in "algorithmic time". The other author proposed a similar view for NN training and showcased how it might lead to more efficient optimization methods [9]. By suggesting a partitioning of the full weight/bias space into a collection of smaller sub-spaces, [9] showed that Koopman training would be an economical optimization technique, if it could accurately model weight/bias evolutions.

Unbeknownst (and in parallel) to us, general work connecting KOT to algorithms was recently explored, and it was conjectured that KOT could be used to speed up NN training [14]. However, [14] focused on solving non-NN numerical problems - we provide a full fledged study of optimizing practically useful NNs. Additionally, we provide a complexity analysis and prove how/when Koopman training is bound to provide computational savings over established methods.

Since our work, two other manuscripts have studied the application of KOT to NNs [15, 16]. In [15], the authors showed how KOT could be used for weight pruning and estimating the number of layers necessary to achieve a certain loss. In [16], the authors investigated how KOT could be used to directly train NNs. Among other differences, a key distinction between [16] and this work is the way in which KOT was implemented. Further, [16] was not able to achieve significant computational cost reductions, while our approach clearly demonstrates how/when it is expected to have a run time complexity better than that of standard optimization algorithms.

There has been little work in machine/deep learning that has leveraged the perspective that NN training is a discrete dynamical system [17, 18]. Our work shares themes with [17–22], all of which took general dynamical systems view points on NN problems, although only [21] looked at the weights directly. However, the focus of [21] was on the macroscopic characteristics of the weight trajectories, while we examined the specific evolution of weights/biases and made predictions using KOT. Finally, recent work has argued that backpropagation is a functor [23]. As Koopman operators are also functors, our work and [23] share themes, but no practical applications were pursued in [23].

## 2 KOT and its connection to NN training

### 2.1 KOT background

KOT is a dynamical systems theory developed by Bernard Koopman in 1931, and then expanded upon by Koopman and John von Neumann in 1932 [24, 25]. In the past two decades, it has had a resurgence due to newly developed analytical and data driven methods [1–8, 26–31].

We begin with the classical perspective on dynamical systems. Let some discrete dynamical system $(\mathcal{M}, t, T)$ be parametrized by the state space variables $[w_1, w_2, w_3, .., w_N] \equiv \mathbf{w} \in \mathcal{M} \subseteq \mathbb{R}^N$, where the evolution of $\mathbf{w}$ over discrete time $t \in \mathbb{Z}$ is governed by the dynamical map $T : \mathcal{M} \to \mathcal{M}$

$$\mathbf{w}(t + 1) = T(\mathbf{w}(t)) \tag{1}$$

Tracking the evolution of $\mathbf{w}$ allows for knowledge of every quantity of interest in the system. To simplify discussions here, we assume the systems under study are autonomous ($T$ is not explicitly dependent on $t$). However, generalizations of KOT to non-autonomous systems are available [32].

KOT takes a different perspective, namely that of studying the observables (state space functions) of $(\mathcal{M}, t, T)$. Let $g : \mathcal{M} \to \mathbb{C}$ be some observable of interest. Here, we assume that $g \in \mathcal{F} = L^2(\mathcal{M}, \rho)$, where $||\rho||_{\mathcal{M}} = \int_{\mathcal{M}} \rho(\mathbf{w}) d\mathbf{w} = 1$ and $\rho$ is a positive, single valued analytic function that supplies the measure for the functional space. The Koopman operator is defined to be the object $U : \mathcal{F} \to \mathcal{F}$, that supplies the following action for $g \in \mathcal{F}$,

$$Ug(\mathbf{w}) = g \circ T(\mathbf{w}) \tag{2}$$

Eq. 2 shows that $U$ is an object that lifts our perspective from the original, possibly nonlinear, dynamical system $(\mathcal{M}, t, T)$ to a new, certainly linear, dynamical system $(\mathcal{F}, t, U)$, thanks to the linearity inherent to the composition $g \circ T$. Extending the definition to describe the associated Koopman operator $U : \mathcal{F}^k \to \mathcal{F}^k$ for some vector observable $\mathbf{g} \equiv [g_1, g_2, ..., g_k]$ is also natural

$$U\mathbf{g}(\mathbf{w}) = \mathbf{g} \circ T(\mathbf{w}) \tag{3}$$

While the linearity of $U$ is an obvious advantage of working under this perspective, difficulties arise from the fact that $U$ is usually an infinite dimensional operator. However, a number of powerful, data-driven methods based on rigorous theory have emerged to accurately approximate the action of $U$ (e.g. [3, 10, 29–31]). The finite section method is one such numerical technique: it constructs an object $\tilde{U}$ to approximate the action of $U$ by using the available time series data [29, 31].

Let us say we have $n$ time series values for $m$ unique observables $g_1, g_2, ..., g_m$. Let the vector $\mathbf{g}(t) = [g_1(t) \; g_2(t) \; ... \; g_m(t)]^\dagger$. Define matrix $F = \begin{bmatrix} \mathbf{g}(1) \; \mathbf{g}(2) \; ..... \; \mathbf{g}(n-1) \end{bmatrix}$ and $F' = \begin{bmatrix} \mathbf{g}(2) \; \mathbf{g}(3) \; ..... \; \mathbf{g}(n) \end{bmatrix}$ ($F'$ is $F$ shifted one time step ahead). The finite section is

$$\tilde{U} = F'F^+ \tag{4}$$

where $F^+ = (F^\dagger F)^{-1} F^\dagger$ is the Moore-Penrose pseudo-inverse and $\dagger$ is the transpose.

### 2.2 NN training as a dynamical system

As noted in Sec. 1, NN training is a discrete dynamical system. Let $\mathbf{w} \equiv [w_1, w_2, ..., w_N] \in \mathcal{M} \subseteq \mathbb{R}^N$ be the $N$ weights/biases of the NN and $T$ be the training algorithm (e.g GD). Then

$$\mathbf{w}(t + 1) = T(\mathbf{w}(t)) \tag{5}$$

describes the training of the NN, where $t$ is the number of completed training iterations. By definition (Eq. 1), this describes a discrete dynamical system parameterized by $\mathbf{w}$, with $t$ serving as the temporal parameter and the training algorithm $T$ supplying the temporal map (also called the cascade or evolution function).

### 2.3 The action of the Koopman operator on NN weights

An important vector observable of $(\mathcal{M}, t, T)$ is the identity observable, $\mathbf{I}(\mathbf{w}) = \mathbf{w}$. By definition,

$$\mathbf{w}(t + 1) = U\mathbf{w}(t) \tag{6}$$

and

$$\mathbf{w}(t + 1) = \mathbf{I}(\mathbf{w}(t + 1)) = \mathbf{I}(T(\mathbf{w}(t))) \tag{7}$$

Necessarily, we have

$$U\mathbf{I}(\mathbf{w}) = \mathbf{I} \circ T(\mathbf{w}) \tag{8}$$

Therefore, knowing the action of the Koopman operator on the identity observable $\mathbf{I}$ allows us to evolve the weights/biases forward exactly like the training algorithm $T$. Precise and sufficient $\mathbf{w}$ evolution data, obtained from previous applications of $T$, could allow an approximation of $U$ using Eq. 4. $\tilde{U}$ could then train that NN further, bypassing further need for the standard training algorithm $T$. This is what is meant by *Koopman training*. In the upcoming sections, we explore whether/when it could be practical, efficient, and accurate for various feedforward, fully connected NNs.

## 3    Implementing Koopman training

Eq. 8 shows that the Koopman training requires time series data for all the NN weights and biases to construct the approximation of the full Koopman operator, $\tilde{U}$, using Eq. 4 (Fig. 1d).

Constructing $\tilde{U}$ using Eq. 4 involves inverting large matrices. Both the construction and usage run time complexity of Koopman training is comprised solely of basic matrix-matrix and matrix-vector operations. We carried out basic run time complexity calculations to assess the expected scaling of run time with the size of the NN and the data matrices $F$ and $F'$ (see Supplemental Material Sec. S1). We found that constructing a single $\tilde{U}$ for the entire NN as a whole comes with costs that would not be preferable in comparison to the standard methods.

This motivated us towards a different approach. The Hartman-Grobman theorem guarantees existence of a neighborhood near hyperbolic fixed point(s), where even nonlinear flows are locally conjugate to a linear flow [33] (recent work has enlarged the space over which this is true to contain the entire basin of attraction [34]). Such a conjugacy also implies structural stability in the region, which in turn implies the capacity of the system to be partitioned into independent sub-systems for easier study and analysis, without being worried about an inordinate loss of accuracy.

We assume a similar setting is possible once NN weights/biases enter regions/basins of local loss minimas, as minimas are fixed points in the dynamical systems perspective. This is to say, we assume that a partitioning of $\tilde{U}$ is possible in some sense, wherein distinct parts (or "patches") precisely evolve some unique subset of NN weights/biases.

We considered several architecturally motivated ways of choosing the partitioning of $\tilde{U}$ (Fig. 1). These include grouping weights/biases by the layers they belong to (Layer Koopman operators), grouping weights/biases by the nodes they connect to (Node Koopman operators), and treating each

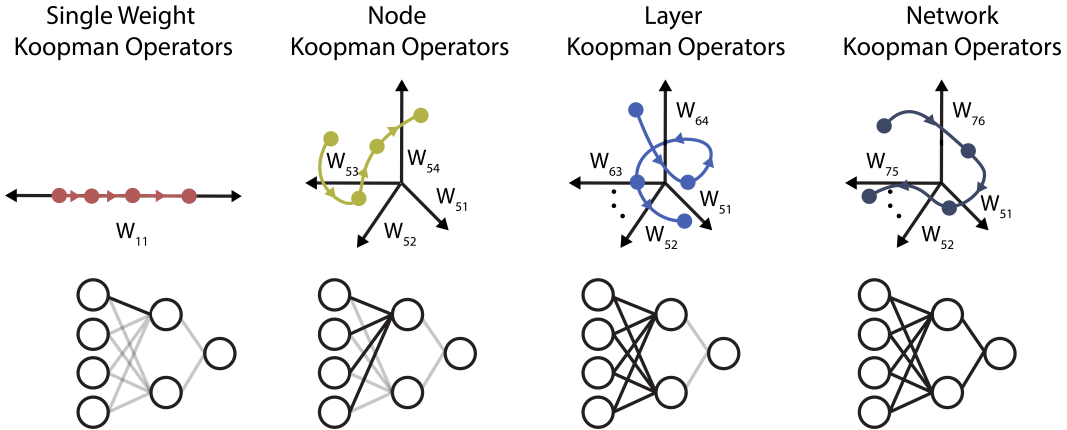

Figure 1: **Examples of possible implementations of Koopman training.** Top, illustration of the trajectory in weight space that the constructed Koopman operators govern. Bottom, illustration of which weights were used to build the Koopman operators (black lines).

weight/bias independently (Single Weight Koopman operators). An intermediate approach between the Node and Single Weight methods ("Quasi-Node") is also possible by dividing the group of weights/biases that connect to a single node into sub-groups and considering those separately.

Note it is abuse of notation to call these partitions Koopman operators, as there is only one Koopman operator for our discrete dynamical system. However, we find these labels conceptually helpful.

Complexity calculations similar to the ones done for the full Network Koopman operator showed that the per iteration complexity associated with training NNs using Node and Single Weight Koopman operators could, in certain cases, be significantly less than that of standard optimizers (see Sec. S1).

While the Single Weight approach is unsurprisingly the fastest, for many NNs the idea of treating each weight/bias independently has obvious problems. Therefore, we expect Single Weight Koopman operators to be very limited in their ability to capture the rich dynamics present in the $\mathbf{w}$ evolution of many NNs. We will primarily focus on training with Node Koopman operators, which balance speed/accuracy to properly deal with the complicated dynamics at hand. However, we note that when the evolution direction of the weights is of more importance than their exact values, Single Weight Koopman operators can be valuable (see Sec. S2 for a simple example).

Pseudo-code for how weight/bias data from standard training iteration $t_1$ to $t_2$ is used to approximate the Node Koopman operators, and how these operators are then used to predict the weight evolution from $t_2 + 1$ to $t_2 + T$, is given in Algorithm 1.

---

**Algorithm 1** Koopman training via Node Koopman operators

---

Let $\mathbf{w}_j^l(t) \equiv [w_{j1}^l(t)\ \ w_{j2}^l(t)\ \ ...\ \ w_{jN}^l(t)\ \ b_j^l(t)]^\dagger$ be weights/bias going **into** node $j$ of layer $l$ after $t$ iterations of training. For each node $(l, j)$, we wish to find $\tilde{U}_j^l$ that satisfies $\mathbf{w}_j^l(t + 1) = \tilde{U}_j^l \mathbf{w}_j^l(t)$.

1. Choose $t_1 < t_2$ and $T > 0$.

2. Train the NN of choice for $t_2$ training iterations and record all $\mathbf{w}_j^l(t)$ from $t_1$ to $t_2$.

3. Define matrices $F = \begin{bmatrix} \mathbf{w}_j^l(t_1)\ \ \mathbf{w}_j^l(t_1 + 1) \cdots \mathbf{w}_j^l(t_2 - 1) \end{bmatrix}$ and
   $F' = [\mathbf{w}_j^l(t_1 + 1)\ \ \mathbf{w}_j^l(t_1 + 2)\ \ \cdots\ \ \mathbf{w}_j^l(t_2)]$.

4. Compute $\tilde{U}_j^l = F'F^+$, where $F^+ = (F^\dagger F)^{-1}F^\dagger$.

5. Use $\tilde{U}_j^l$ to predict the evolution of $\mathbf{w}_j^l$ from $t_2$ to $t_2 + T$.

---

## 4 Koopman training a Neural Network Differential Equation Solver

We began by applying Koopman training to a feedforward, fully connected, deep NN differential equation (DE) solver designed for studying Hamiltonian systems [35, 36]. The NN used unsupervised learning to generate smooth functional approximations for the expected solutions of nonlinear and chaotic dynamical systems. It was demonstrated to be significantly more accurate than traditional numerical integrators at an equivalent level of discretization. In general, NN DE solvers are a rapidly emerging class of tools for attacking complex systems, as they have been found to have benefits that traditional methods do not. This is partly because they can embed important features of nominally high dimensional systems in low dimensional settings and partly because they can leverage the immense parallelization capacities of modern GPUs. We note that NN DE solvers are in no way optimized, or otherwise specially suited, for Koopman training.

A schematic of the NN DE solver is shown in Fig. 2a. For ease of implementation, we modified it to have sigmoidal activation functions, a fixed batch (making the optimization non-stochastic), a learning rate with a weak training step dependent decay, and trained with various optimizers via backpropagation (see Sec. S3 for more details). Because different optimizers induce different $\mathbf{w}$ dynamics, we tried three different popular ones using the PyTorch environment [37]: Adam [38], Adagrad [39], and Adadelta [40]. Because the three were similarly successful, we focus on the Adadelta results below. See Table 1 and Figs. S2 and S3 for the other choices.

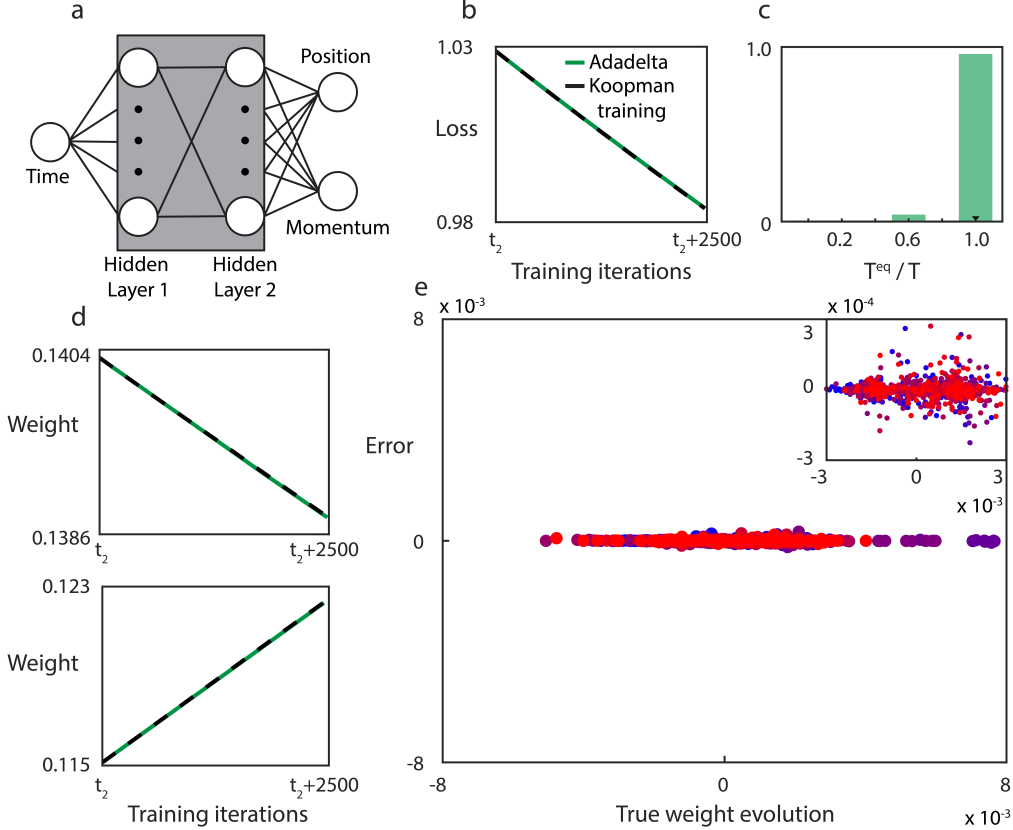

Figure 2: **Koopman training an NN DE solver that used Adadelta.** (a) Schematic of the NN DE solver [35, 36]. (b) Koopman trained network loss (black dashed line) compared to the loss of the corresponding Adadelta trained network over $T$ training iterations (green solid line). (c) Distribution of $T^{\text{eq}}$ for all the randomly initialized NNs. (d) Examples of evolutions of individual weights/biases for a Koopman trained NN, as compared to the corresponding evolutions for the Adadelta trained NN. (e) Comparison of the error in all the individual weight/bias predictions, as a function of the amount the true (Adadelta) weights/biases evolved for the 10 networks with the smallest mean error. The bluer the color, the smaller that mean error. *Inset:* Zoomed in around (0, 0) of the main figure.

## 4.1 Koopman training loss closely follows the evolution of the standard optimizer loss

Like any NN optimization technique, the value of Koopman training is in the loss performance it provides. To benchmark Koopman training for iterations $t_2 < t < t_2 + T$, we tracked its loss performance and compared it against the loss performance of standard optimizers for the same NN over the same training iterations. An example is shown in Fig. 2b. We see that the loss evolution via Koopman training very closely matches that of Adadelta. To precisely quantify this across the NNs that we randomly initialized and investigated, we computed the number of Adadelta iterations needed to reach the same loss as was achieved after $T$ Koopman training steps (for a fixed $T$). We call this $T^{\text{eq}}$ ("eq" stands for equivalent). If $T^{\text{eq}}/T > 0$, Koopman training was able to lower the NN loss from where it was when Koopman training started ($t_2$), and we consider it a success. The closer to $T^{\text{eq}}/T$ is to 1, the closer Koopman training approximates Adadelta, in terms of the loss performance per iteration. Distribution of $T^{\text{eq}}/T$ for all the randomly initialized NNs is in Fig. 2c. All but one NN had $T^{\text{eq}}/T \approx 1$, showing that Koopman training was indeed generally able to closely approximate the loss of the Adadelta trained NN.

## 4.2 Koopman training accurately predicts the true weight/bias evolution

While it is encouraging to see that the loss evolution of the Koopman trained network closely approximates that of Adadelta, the theory that was laid out in Sec. 2 was developed in terms of the

NN weights/biases. How well did Koopman training predict their evolution? Were the most dynamic weights/biases accurately predicted?

Examples of individual weight/bias evolution under Adadelta and Koopman training are shown in Fig. 2d. These examples show that the Koopman trained weights/biases track the Adadelta trained weights/biases well across the range of time Koopman training was applied for. For Adadelta, we found that, by the time we applied Koopman training, the weight/bias trajectories were largely straight lines. However, note that Koopman training is *not* finding linear fits for the **w** evolution, it is identifying the underlying dynamics, whatever their nature may be. This is made apparent in Sec. 5, where we successfully Koopman train when the weight/bias trajectories were not straight lines.

To quantify the accuracy of the Koopman training predictions across all the NNs we examined, we looked at the difference between the Koopman trained weights/biases and the "true" Adadelta weights/biases (i.e. the error). We plot this as a function of the amount the Adadelta weights/biases evolved from $t_2$ to $t_2 + T$ (i.e. the "true" weight evolution) for the ten NNs with the smallest average error magnitude (Fig. 2e). This analysis revealed two things. First, the error was, on average, at least two orders of magnitude smaller than the amount of "true" weight/bias evolution (the median ratio between them is 0.0044). Second, the most dynamic weights/biases (corresponding to the points on the farthest ends of the x-axis) were among the best predicted.

In conclusion, Koopman training was indeed able to accurately predict weight/bias evolution over a non-trivial length of training time. A single Koopman training step was - at the weight/bias level - approximating the action of a single Adadelta training step. This clearly demonstrates how/why Koopman training can provide loss performance similar to that of standard optimizers like Adadelta.

## 4.3 Koopman training is significantly faster

As discussed in the Sec. 1, a principle motivation for this work was to see whether KOT could help reduce the computational costs associated with NN optimization. To investigate this, we compared the amount of time it took to do $T$ Koopman training steps with the amount of time it took to do $T^{\text{eq}}$ Adadelta training steps (see Sec. S3 for more details). As reported in Table 1, we found that Koopman training was, on average, two orders of magnitude faster over the range of training it was applied to. We note that this was compared to PyTorch's highly optimized implementations of the standard training methods [37–40]. Additionally, we didn't implement any parallelization options. Koopman training solely involves matrix-matrix and matrix-vector operations - it is *significantly* more amenable to parallelization than Adadelta and other standard training methods. Therefore, we expect our results to be even more significant when parallelization is implemented.

Table 1: **NN DE solver Koopman training results.** Median values are reported for $T^{\text{eq}}/T$ and speedup. Additional information and other examples are detailed in Sec. S3.

| Architecture | Optimizer | Success | $T^{\text{eq}}/T$ | Speedup |
|---|---|---|---|---|
| 1:10:10:2 | Adam | 84% | 0.85 | 98x |
| 1:10:10:2 | Adagrad | 92% | 1.00 | 132x |
| 1:10:10:2 | Adadelta | 100% | 1.00 | 141x |

## 5 Koopman training an MNIST network

Powerful KOT inspired techniques exist for studying a very wide variety of dynamical systems (even random dynamical systems [41]). As discussed in Sec. 2, appropriate variations of Koopman training should be possible/successful for networks with different architectures, objectives, optimizers, and activation functions (all of which lead to different kinds of **w** dynamics). To test the versatility/power of Koopman training, we applied our simple approach to a three-layer feedforward, fully connected NN with ReLU activation functions. This NN was trained on the MNIST data set via stochastic Adadelta (Fig. 3a - see Sec. S4 for more details). Finally, the Koopman training implementation used the Quasi-Node method for the weights/biases belonging to the first layer, due to it being significantly larger than the other layers. Other weights/biases were handled using Node Koopman operators.

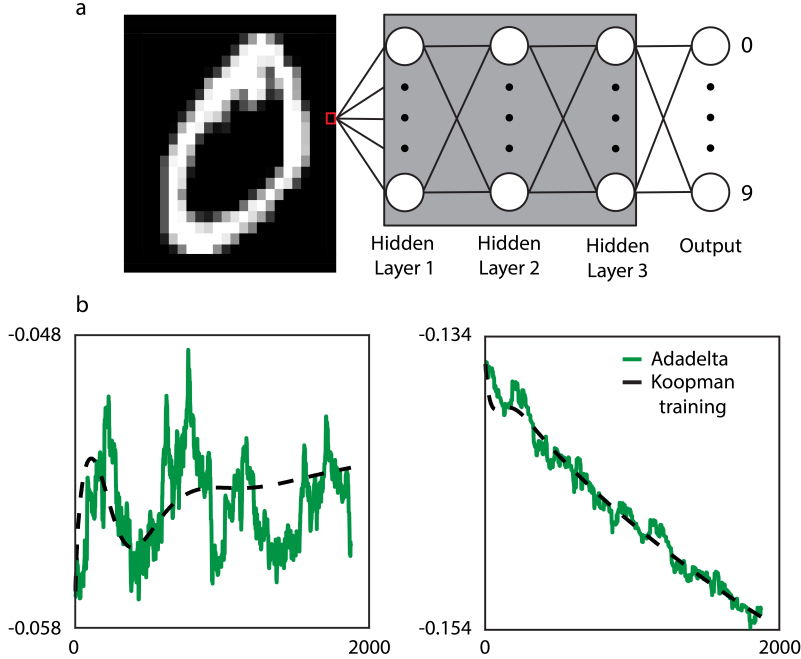

Figure 3: **Koopman training a feedforward, fully connected NN trained on MNIST using stochastic Adadelta.** (a) Schematic of the feedforward, fully connected NN that was trained on MNIST. (b) Example of weight/bias trajectories of the Koopman trained NN (black dashed line) and the corresponding Adadelta trained NN (green solid line).

The NN made a distinction between training and validation loss. This, in addition to the stochastic weight/bias evolution meant that some of the metrics used in Sec. 4 were no longer informative (see Sec. S4 for an adjusted definition of $T^{eq}$). For instance, just because the Koopman trained weights/biases differed from the true weights/biases did not mean that Koopman training was doing a poor job at properly training the NNs. In general, we found that Koopman training tracked the underlying trends of the weights well (Fig. 3b). We emphasize that these Koopman trained weights/biases took nonlinear forms, showing that KOT can sufficiently describe the nonlinear dynamics in weight/bias space. As with the NN DE solver example, accurate tracking of the underlying weight/bias dynamics led to good loss performance. Koopman training was 100% successful, had $T^{eq}/T \sim 0.97$, and was 15x faster over the range of training iterations it was implemented upon.

These results support the idea that Koopman training can be successfully applied to a range of feedforward, fully connected NNs with different architectures, objectives, activation functions, and optimization algorithms (including stochastic ones).

Table 2: **MNIST NN Koopman training results.** Median values are reported for $T^{eq}/T$ and speedup. Additional details are provided in Sec. S4.

| Architecture | Optimizer | Success | $T^{eq}/T$ | Speedup |
|---|---|---|---|---|
| 784:20:20:20:10 | Adadelta | 100% | 0.97 | 15x |

## 6  Conclusion

In this work, we demonstrated the first application of Koopman operator theory for the optimization of neural networks. Computational complexity calculations showed that Koopman training via the full Network Koopman operators was not expected to be faster than standard NN training methods. This led us to an architecturally motivated "partitioning" of the full Network Koopman operator into

smaller pieces. We argued and verified that this approach would work when the NN weights/biases are near regions of local minimas (fixed points in the dynamical systems perspective). Through several examples of feedforward, fully connected NNs, we confirmed that Koopman training was indeed able to correctly evolve individual weights/biases over a non-trivial range of training iterations (Fig. 2d, 2e, and Fig. 3b). That is, Koopman training well approximated the action of standard training algorithms. Consequently, the Koopman trained NNs had comparable loss performance to the standard trained NNs over the same range of training time (Fig. 2b and 2c). As our complexity analysis suggested, Koopman training was orders of magnitude faster (Tables 1 and 2), making Koopman training an attractive alternative to standard optimization techniques. We note that our simple implementation was compared against the highly optimized PyTorch environment [37].

Given the novelty of Koopman training, our work is accompanied by many unaddressed questions that shall be the subject of future work in this exciting new intersection between dynamical systems and NN/optimization theory. One particularly important question is how to choose when we start collecting weight/bias data (i.e. $t_1 = ?$) and when we turn on Koopman training (i.e. $t_2 = ?$). For now, we used a 'one size fits all" approach by fixing $t_1$ and $t_2$ and ignoring any variations in where a given NN was in the $\mathbf{w}$ evolution. As each random initialization of an NN will lead to different trajectories with radically different dynamics (different number of iterations to reach the basins of attraction, different local minimas, etc.), refined implementations of Koopman training could adaptively determine $t_1$ and $t_2$. KOT itself can be used to identify when the system has reached a basin of attraction: having all the eigenvalues of the Koopman operator lying on the unit circle in the complex plane is a robust indicator [3].

Another key question relates to whether there are "smarter" ways of implementing the partitioning of the full Network Koopman operator? Our method relied on an architecturally motivated partitioning, but there are various other schemata that could be considered. Whether/how additional information about individual weight/bias trajectories could be used to capture more details about the dynamics in each patch remains an interesting open question.

Additionally, the question of how much Koopman training can be used to speedup standard training is also open. Our implementation of Koopman training relied on standard, but simple, KOT methods. Indeed, there are many other methods that could increase the speed advantage. For instance, implementing the Cholesky method in computing the pseudo-inverse of Eq. 4 has been found to significantly decrease the run time of certain KOT algorithms [42]. Additionally, there exist popular KOT algorithms, such as variants of dynamic mode decomposition [2, 29], that allow for more direct identification of the fixed points. While some of these approaches were used in other recent works involving KOT and NNs [15, 16], they have not been used in conjunction with our partitioning approach. This was a key insight that separates our paper from the others and is what, under certain assumptions, leads to a faster run time than standard training. It will be interesting to see how applying such methods to our implementation of Koopman training affects the speedup.

Finally, as noted in Sec. 4.3, Koopman training is significantly more amenable to parallelization than standard training methods, because it involves purely matrix-matrix and matrix-vector operations. Parallelizing Koopman training therefore promises even greater advantages in speed. KOT is an exciting, powerful, and growing theory, and we hope that this work illustrates its potential as a tool for those studying and training NNs.

## Broader Impact

Koopman training is an exciting, novel approach to optimizing neural networks. We believe it can be generally used to reduce the computational costs consumed during the training phase, a claim our complexity calculations and numerical examples support. These savings, which more advanced methods of Koopman training should increase, may translate into lowering the monetary costs, time, and energy needed to optimize neural networks and allow those without significant funding and computing power (e.g. researchers at non-R1 universities and small companies) to train neural networks to higher utility. As NNs have become a staple in an ever growing number of academic/commercial fields, Koopman training has the potential for wide impact. Additionally, our work should generalize to other types of complicated, iterative optimization dependent problems.

Finally, this paper is among a growing minority of work that highlights the power viewing neural networks from a dynamical systems perspective can bring ([9, 15, 16, 43]). We hope that this work inspires future exchanges between the neural network and dynamical systems communities.

## Acknowledgments and Disclosure of Funding

We thank Mr. Arvi Gjoka (NYU) for his feedback on neural networks and machine learning. We thank Dr. Felix Dietrich (TMU), Prof. Yannis Kevrekidis (JHU), and Prof. Igor Mezic (UCSB) for discussions on Koopman operator theory. We also thank Prof. Julio Castrillon (BU), Prof. Mark Kon (BU), Dr. N. Benjamin Erichson (UC Berkeley), Mr. Chris Ick (NYU), and Mr. Cory Brown (UCSB) for their constructive conversations. Finally, we thank the anonymous *NeurIPS* reviewers whose comments were very helpful in making this work clearer and leading us to deeper results.

No third party funding/resources were used to support this work. The authors declare no competing interests.

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
