[Supplementary Material]

# Optimizing Neural Networks via Koopman Operator Theory (Supplemental Material)

**Akshunna S. Dogra**[*]
John A. Paulson School of
Engineering and Applied Sciences
Harvard University
Cambridge, MA 02138
asdpsn@gmail.com

**William T. Redman**[*]
Interdepartmental Graduate Program in
Dynamical Neuroscience
University of California, Santa Barbara
Santa Barbara, CA 93106
wredman@ucsb.edu

## S1  Managing the complexity of Koopman training

As discussed in Sec. 3 of the main text, the computational complexity of Koopman training is dependent on the approach we take to estimate the action of the Koopman operator on the weights and biases $\mathbf{w}$. We will develop and deepen the idea that our implementation of Koopman training, based on the finite section method, can accurately "mimic" standard NN training, while providing computational cost savings. There are two costs associated with our implementation of Koopman training: the construction cost of $\tilde{U}$ and the per iteration cost of using $\tilde{U}$ to drive the $\mathbf{w}$ evolution.

Let a feed-forward, fully connected, deep NN have $n-1$ hidden layers, input/output dimensions $n$, and constant width $n$, giving us a total of $\sim n^3$ weights/biases. We assume that the various possible approaches to Koopman training discussed in Sec. 3 (Fig. 1) are built using $\mathbf{w}$ data from $k$ iterations. We assume that both standard training and Koopman training use simple matrix computation methods. Lastly, we assume the batch size is $j$. Thus, the per iteration computational cost of any standard backpropagation algorithm is at least $\mathcal{O}(jn^3)$ [1]. Further, the operations needed for computing every activation function, for estimating the loss $L(\mathbf{w})$, for estimating $\frac{\partial L}{\partial \mathbf{w}}$, etc., can lead to reasonably large coefficients. We note that none of these factors are relevant for Koopman training.

Let us first obtain the run time complexity of constructing a single $\tilde{U}$ for some arbitrary $m$ dimensional observable, using data from $k$ training iterations (usually, $k \gg m$). The finite section method, Eq. 4, implies the run time complexity would be $\mathcal{O}(\max\{m^3, km^2\})$. The per iteration cost of using $\tilde{U}$ to drive the evolution is simply $\mathcal{O}(m^2)$, since it involves multiplying a $m \times m$ matrix by a $m$-dimensional vector.

The simple considerations above allow us to make useful estimates of the run time complexity associated with various Koopman training implementations.

1. *Single Weight KOs*: This approach gives $n^3$ unique scalar observables ($m = 1$), each accounting for an individual weight/bias of our NN. This approach effectively takes the $n^3$ – dimensional evolution associated with $\mathbf{w}$ and partitions it into $n^3$, independent, 1 – dimensional evolutions.
   **Construction complexity**: $\mathcal{O}(kn^3)$. **Per iteration complexity**: $\mathcal{O}(n^3)$.

2. *Node KOs*: This approach gives $n^2$ unique vector observables ($m = n$), each containing all the weights/biases that connect individual layers to one unit of the next layer. This approach effectively takes the $n^3$ – dimensional evolution associated with $\mathbf{w}$ and partitions it into $n^2$, independent, $n$ – dimensional evolutions.
   **Construction complexity**: $\mathcal{O}(\max\{kn^4, n^5\})$. **Per iteration complexity**: $\mathcal{O}(n^4)$.

---

[*]The authors contributed equally

- *Quasi-Node KOs*: This approach gives $\frac{n^3}{q}$ unique vector observables ($1 < m = q < n$). It is the compromise between the computational efficiency of the *Single Weight* approach and the relatively better accuracy of the *Node* approach. This is achieved by creating unique, smaller, independent $q$ – sized sub-groupings of the weights/biases going to a single node - it limits to *Single Weight* when $q = 1$ and *Node* when $q = n$. **Construction complexity**: $\mathcal{O}(\max\{kqn^3, n^3q^2\})$. **Per iteration complexity**: $\mathcal{O}(qn^3)$.

3. *Layer KOs*: This approach gives $n$ unique vector observables ($m = n^2$), each accounting for all the weights/biases connecting one layer to the next. This approach effectively takes the $n^3$ – dimensional evolution associated with $\mathbf{w}$ and partitions it into $n$, independent, $n^2$ – dimensional evolutions.
   **Construction complexity**: $\mathcal{O}(\max\{kn^5, n^7\})$. **Per iteration complexity**: $\mathcal{O}(n^5)$.

4. *Network KOs*: This approach gives a single vector observable ($m = n^3$), accounting for every weight/bias in the NN. This approach directly studies the full $n^3$ – dimensional evolution associated with $\mathbf{w}$.
   **Construction complexity**: $\mathcal{O}(\max\{kn^6, n^9\})$. **Per iteration complexity**: $\mathcal{O}(n^6)$.

Approaches 1 – 3 are a way of partitioning/decomposing the full complex dynamics associated with $\mathbf{w}$ into multiple, independent, relatively simpler sub-dynamics. We then approximate the associated Koopman operator(s) and evolve each partition separately from the others. This may result in some loss of accuracy, but also provides a significant computational speed up. In Sec. 3, we discussed when we think this "patching" approach should give small errors. It is quite normal for the batch size $j \gtrsim n_N$, where $n_N^3$ is the total number of weights associated with a NN (in our example, $n_N^3 \sim n*n*n = n^3$). This makes both the Single Weight and Node Koopman operator techniques viable alternatives to standard NN training, especially since Koopman training complexity does not come with large constant coefficients. In the NN DE solver example we considered in Sec. 4 of the main text, $j >> n_N$, which led us to expect significant speedups when compared to standard NN training. This was empirically verified by our results, where our simple implementation of Koopman training turned out to be about 100 times faster than the standard optimizers used in Pytorch.

Note that other, possibly "smarter", partitioning choices may exist. We were guided by the intuition that it might be advantageous to group weights by their sources/sinks in the architecture - it is possible that the best partitioning choice(s) are completely independent of the NN architecture and strictly dependent on the size of each partition instead (or vice versa or a mix of the two or neither). A deeper look into partitioning choices will be the focus of our future work.

In terms of accuracy, Node Koopman operators are at best equivalent to knowledge of the $n$ most relevant spectral parameters of the true Koopman operator (as a Koopman operator is a linear, albeit infinite dimensional operator). This might not be enough in a highly sensitive region of training (such as near weight initialization). However, once the $\mathbf{w}$ evolution enters the basin of a local minima (a fixed point from the dynamical perspective), Node KOs can be more than adequate. This was empirically verified by our results.

Similarly, in situations where accurate tracking of the weights is less important than identifying long term trends, the Single Weight approach is an even more powerful alternative to standard NN training. An example is discussed in the next section.

## S2 Koopman training a perceptron: benefits of the Single Weight approach

### S2.1 Motivation

As mentioned in Sec. 3 of the main text, and developed further in the run time complexity analysis in Sec. S1, the Single Weight approach to patching/partitioning has the least run-time complexity. However, by its very nature it considers each weight/bias independently, a construction that makes little sense when $\mathbf{w}$ evolution has to be estimated as accurately as possible. As we illustrate here, there are cases where this approach can be a reasonable and considerably more efficient choice. In the example ahead, identifying the long-term dynamical trends of weight/bias evolution is more important than the accurate tracking of the weights/biases themselves. While a simple toy problem, we imagine there are other more important cases where the intuition provided here could be useful.

Figure S1: **Koopman training a perceptron.** (a) A schematic of the four unit input, two unit output perceptron that we used. Learning the task amounted to strengthening the "relevant" weights (thick lines) and weakening the "irrelevant" weights (thin lines). (b) Typical example of the evolution of the relevant (top) and the irrelevant (bottom) weights using just perceptron training and using different Koopman training procedures. (c) Schematic of the three different methods used for implementing Koopman training. (d) Percent error during training, averaged across 1000 simulations. A sliding window, of width 50 training steps, was used to smooth the curves. Shaded area represents mean $\pm$ standard error of mean (SEM).

## S2.2 Perceptron

We trained a four unit input, two unit output perceptron. Each of the output units learned to perform a logical OR on one half of the input units (Fig. S1a). That is, the first (second) output unit was trained to be active iff at least one of the first (last) two input units was active. Over the course of the training, the NN learned to weaken the "irrelevant" weights and strengthen the "relevant" ones (Fig. S1b black lines - widths denoting relevance). All parameters used for the perceptron and its training are in given in Table S1.

For this NN, the training at each time step is determined by the NN weights $\mathbf{w}$, the input $\mathbf{x}$, and the target output $\mathbf{y}$. The actual value of $i^{\text{th}}$ output unit, $\hat{y}_i$, is given by

$$\hat{y}_i = f\big(\sum_j w_{ij}x_j\big) = f\left(\mathbf{w}_i \cdot \mathbf{x}\right) \tag{S1}$$

where $f$ is a nonlinear function. In this case, $f$ was the step function

$$f(\mathbf{w}_i \cdot \mathbf{x}) = \begin{cases} 1 & \text{if } \mathbf{w}_i \cdot \mathbf{x} > 1, \\ 0 & \text{otherwise} \end{cases} \tag{S2}$$

Table S1: The parameter values used for the perceptron in Fig. S1.

| Parameter | Value |
|---|---|
| Learning rate ($\eta$) | 0.005 |
| Number of input units | 4 |
| Number of output units | 2 |
| Probability of input unit being 1 | $1/4$ |
| Number of training steps | 1000 |
| Number of separate simulations | 1000 |
| Initial weight domain | $[0.5, 1]$ |

The perceptron learning rule is given by

$$\Delta \mathbf{w} = -\eta \, \mathbf{e} \qquad \mathbf{e} = \hat{\mathbf{y}} - \mathbf{y} \cdot \mathbf{x}^T \qquad (S3)$$

where $\eta$ is the learning rate and $\mathbf{e}$ is the error [2].

The results of applying each of the Koopman training approaches (Fig. S1c) to the perceptron after 100 training steps, as compared to training using only the perceptron rule, are shown in Fig. S1d. All three implementations of Koopman training led to a reduction in the number of training steps needed to reach asymptotic performance, however the Single Weight Koopman training approach felt this advantage the strongest. There are two major reasons for this, as seen in Fig. S1b. First, the Koopman evolved weights grow/decay faster. In this simple example, a perfect network has relevant weights greater than one and irrelevant weights less than one-half. Because continued growth/decay has no effect on performance, Koopman training has the network reach, and stay at, the optimal state sooner. Second, whereas the perceptron rule requires constant data to train, some of which can lead to no change in the weights, Koopman training does not, and therefore continually pushes the weights towards their optimal states. Because Single Weight Koopman operators cause exponential growth/decay, they lead to the fastest approach to asymptotic performance. Thus, Single Weight Koopman training is a powerful tool when identifying the individual long-term dynamical trends of weight evolution is more important than the accurate tracking of the weights themselves.

### S2.3 Scaling up the size of the perceptron - a cautionary note

For completeness, we include the following point (unrelated to Single Weight Koopman training). We found that if we increased the number of input units, $n_1$, while setting the probability that a given input unit would be active to $1/n_1$, we didn't get as strong results as we had in Fig. 1 (except for the Single Weight Koopman training, where the results are comparable). To see why this is the case, note that the perceptron leaning rule, Eq. S3, only updates the weights of units that are active. As $n_1$ increases, the expected proportion of relevant/irrelevant weights that change together will decrease. And because only a small amount of data is used to construct the Koopman operators, the "correct" dynamics (which would be that all the relevant weights increase and all the irrelevant weights decrease) is poorly approximated. This leads to poor Koopman training. By more carefully choosing the data used to train the perceptron (namely, increasing the number of input units in the same half that are active together), we were able to recover the results from Fig. 1 for larger input sizes.

We choose to include this point because it illustrates the limitations of the perceptron on the task we selected, highlights the importance of what data is used for training (even for Koopman training), and serves as a reason for caution when attempting to use Koopman training on networks whose weights can/do change in an uncorrelated way. Additionally, it highlights another area where Single Weight Koopman training can avoid some of the technicalities that come with the other approaches.

## S3   Koopman training a NN DE solver

The code for the feedforward, fully connected NN DE solver explored in the main text (Sec. 4 and Fig. 2) was retrieved from [3]. Custom changes were made by the authors, so that weight/bias values could be saved and Koopman training weights/bias could be directly input into the NN. The architectures of all the numerical experiments are given in Table 1 of the main text. All weight/bias data was saved

Figure S2: **Koopman training an NN DE solver that used Adam.** Subfigures are the same as Fig. 2 of the main text.

in double precision (unlike the original code, where the weights/biases were in single precision) and then manipulated in Matlab 2018b using custom code. All computing was done serially on a Dell work station with 3.4 GHz Intel Core i7-6700 and 32 GB RAM.

## S3.1 Adam

The parameters for the NN DE solver using Adam in Fig. S2 and Table 1 are given in Table S2. Any values not specified are the same as was used in [4].

Table S2: NN DE Adam

| Parameter | Value |
|---|---|
| Learning rate $(\eta(t))$ | $8/(1000 + t)$ |
| Betas | $(0.999, 0.9999)$ |
| Noise | 0 |
| Number of training steps | 60000 |
| $t_1$ | 35000 |
| $t_2$ | 45000 |
| Number of separate simulations | 25 |
| Activation function | sigmoid |

## S3.2 Adagrad

The parameters for the NN DE solver using Adagrad in Fig. S3 and Table 1 are given in Table S3.

Table S3:  NN DE Adagrad

| Parameter | Value |
|---|---:|
| Learning rate ($\eta(t)$) | $8/(1000 + t)$ |
| Noise | 0 |
| Number of training steps | 60000 |
| $t_1$ | 35000 |
| $t_2$ | 45000 |
| Number of separate simulations | 25 |
| Activation function | sigmoid |

Figure S3: **Koopman training an NN DE solver that used Adagrad.** Subfigures are the same as Fig. 2 of the main text.

## S3.3 Adadelta

The parameter used for the NN DE solver using Adadelta in Fig. 2 and Table 1 are given in Table S4.

Table S4: NN DE Adadelta

| Parameter | Value |
|---|---|
| Learning rate ($\eta(t)$) | $8/(1000+t)$ |
| Rho | 0.999 |
| Noise | 0 |
| Number of training steps | 60000 |
| $t_1$ | 35000 |
| $t_2$ | 45000 |
| Number of separate simulations | 25 |
| Activation function | sigmoid |

## S4   MNIST

The code for the feedforward, fully connected NN that was trained on the MNIST data set was taken from [5] and modified so that the layers were feedforward and not convolutional. The parameters are given in Table S5. We emphasize, the weight and bias data used to build the Koopman operators was from the start of Epoch 3 ($t_1$) up until the end of Epoch 5 ($t_2$), each epoch had 938 iterations.

MNIST NN, unlike the NN DE solver, implemented stochastic Adadelta and had weights/biases change without evaluating the validation loss at every iteration (i.e. had epoch structure). Hence, $T^{\text{eq}}$, as defined in the main text, could only take integer values and had a small range. This made evaluating the performance of Koopman training tricky and unclear. Therefore, we modified our definition of $T^{\text{eq}}$ in the following way.

Let $\{\ell_i\}$ be the validation loss of the Adadelta trained NN after $t_2$ and let $\ell_T^{\text{KO}}$ be the loss of the Koopman trained network after $T$ time steps. Then $T^{\text{eq}}$ was computed as the sum of two parts

$$T^{\text{eq}} = Q + R \tag{S4}$$

where $Q = \max_i\{i|\ell_i > \ell_T^{\text{KO}}\}$ and $R = (\ell_Q - \ell_T^{\text{KO}})/(\ell_Q - \ell_{Q+1})$. Note that $Q$ is the original definition of $T^{\text{eq}}$ and $R$ tells us how close the loss of the Koopman trained NN is to the loss of the Adadelta trained NN at the next epoch.

Let $\{\tau_i\}$ be the run time of each epoch after $t_2$. The equivalent run time (used to compute the speedup reported in Table 2 of the main text) was defined as

$$\sum_{i=1}^{Q} \tau_i + R\tau_{Q+1} \tag{S5}$$

Table S5: The parameter values used for the feedforward, fully connected NN trained on MNIST using Adadelta in Fig. 3 and Table 2. Any values not specified are the same as were used in [5].

| Parameter | Value |
|---|---|
| Learning rate | 1.0 |
| Gamma | 0.7 |
| Batch size (training) | 64 |
| Batch size (testing) | 1000 |
| Number of training epochs | 10 epochs |
| $t_1$ | Epoch 3 |
| $t_2$ | Epoch 5 |
| $T$ | 2 Epochs |
| $q$ | 157 |
| Number of separate simulations | 25 |
| Activation function | ReLU |