[Reviews · NeurIPS 2020]

Review 1

Summary and Contributions: This paper provides a new perspective on the training procedure of NNs based on the Koopman operator theory. By considering the weights of NNs as states of a dynamical system, the linear Koopman model can be utilized to characterize and speed up the training of NN.

Strengths: This paper provides a novel approach for Koopman analysis of NN training. Although the further investigations on theory, algorithm and scalability is required (see below), this work is a significant step forward in understanding and improving GD based learning approaches.

Weaknesses: 1) In applications related to dynamical systems, accurate Koopman models can be obtained only if the trajectory data can well cover the state space. Due to this fact, it seems that the Koopman training can only refine the solutions after GD found some satisfying ones. This problem might not be serious for simple examples, but experiments on large-size learning problems are required. 2) Considering the stochasticity of NN training, the method is in fact developed based on stochastic Koopman operator instead of Koopman operator. 3) In applications related to dynamical systems, the linear Koopman model approximates nonlinear systems by using a large number of nonlinear observables, but this paper only considers the linear observable. Can such a model well capture the essential part of the system? === Update after rebuttal: In my opinion, this paper in fact perform "extrapolation" in some sense and try to "predict" the stable point of SGD, so the performance on practical experiments is still questionable for me. After reading all the comments and the rebuttal, I decide to slighly descrease the score. It provides an interesting direction for deep learning, but is not well developed for now.

Correctness: Most of the claims and method are correct. One problem comes from the Koopman training of deep NN, where the learning rate is time-varying and therefore the corresponding of dynamics of training is also time-varying. Further discussions are required.

Clarity: The paper is well written, but the clarity can be improved. A detailed pseudocode can help readers to understand the Koopman training algorithm more easily.

Relation to Prior Work: It would be better if the authors can compare their method with second-order training algorithms (including Newton/Quasi-Newton/Diagonal Newton). The latter ones also utilize locally linear models, and involve matrix estimation and product.

Reproducibility: Yes

Additional Feedback:


Review 2

Summary and Contributions: This paper proposes apply Koopman operator theory into the training of modern NN. The experimental results demonstrate the efficiency of the proposed method when training a shallow fully connected network. 10x training speed-up is promising.

Strengths: The paper is the first to bridge KO to NN training. The experiments are promising in the sense of 10x training speed-up.

Weaknesses: The proposed training method is not clearly stated. As the Koopman training method is only introduced for simple examples, how to implement Koopman training to general NN structures and why Koopman training can beat GD are not clearly stated. The setting of the experiments is not general and the potential to extend the method to general machine learning task or physical task is not discussed.

Correctness: Yes

Clarity: Not very good. I suggest to re-organize the paper sections, especially leverage subsections to separate Section 2 and introduce the background and your proposed method separately. You may also leverage algorithmic frames to better illustrate the proposed training method. If the introduced method at the end of section 2 is not closed related to other contents, it's better to move it to future work section at the end of the paper.

Relation to Prior Work: Yes

Reproducibility: Yes

Additional Feedback: 1. What is the benefit of regarding NN training as KOT? Does it the convexity of the operator "U"? However, NN model is non-convex in nature, why KOT can accelerate NN training is not clear to me. 2. As section 3 only introduced the Koopman training for simple examples, it is not clear to me how to implement Koopman training to general deep fully connected NNs. Can the proposed method be extended to other NN structures such as ResNet, Transformer? 3. Is the method specially designed for studying Hamiltonian systems? I think that the result on some famous tasks (e.g. image classification) would be more persuasive. 4. "The simplest approach is to consider each weight’s trajectory independent of all the others." This assumption is not satisfied for MLP.


Review 3

Summary and Contributions: This paper advocates for the usefulness of Koopman Operator Theory (KOT) in the training and analysis of neural networks. Koopman operator theory is likely to be unfamiliar to many in the Neurips community (and indeed was to me). At a high-level, it is an approach that (through appropriate transforms) allows us to shift our perspective of a non-linear dynamical system into a different isomorphic system that has linear dynamics. Whilst forming this mapping might be intractable in many cases, there exist data-based approximation methods available. In particular, the paper argues and demonstrates that by taking appropriate observations of learning-dynamics/state-evolution during the initial stages of neural network training, we are able to build a sufficiently accurate estimate of a Koopman operator so that we can then apply this operator (instead of traditional gradient based optimization) to obtain estimates for the trained parameters of a network. This ability potentially leads to computational savings, as well as other advantages. The authors demonstrate these ideas on two small test-cases. The first, a four-input + two-output perceptron, required to learn two independent OR functions. The second, a very small, two hidden layer neural net (architecture layer sizes: 1 -> 4 -> 4 > 2) trained to as a solver for Hamiltonian systems. In both cases the proposed KOT method is shown to be capable of training the network models comparably well to standard gradient descent and with savings in terms of computational burden.

Strengths: I found this paper to be interesting, and with useful insights that could eventually have considerable relevance to and impact on the Neurips community. KOP was novel to me, though in reviewing the referenced related literature I see that it has been discussed in the context of machine learning already and so the main contribution of this paper seems to be demonstrating that it can be used in practice. Despite the very small scale of the demonstrations presented, this paper could help catalyse interest and progress in a subfield that may hold substation potential. To the best of my knowledge, the theoretical claims are sound.

Weaknesses: For me, one of the biggest limitations of this work is the tiny scale of the evaluations. Both in terms of the size of the networks considered, but also in the lack of real diversity in problem types explored. Were these lacunae addressed, I think it would help improve the work substantially and would also make the potential value proposition clearer to other practitioners and thereby help increase that chance this work/direction has the significance of its potential impact explored sooner by a larger number of researchers. The perceptron results are perhaps interesting theoretically, but given the nature of the problem treated have limited practical value. The network considered for Hamiltonian dynamics is also incredibly small -- and may be a problem that is unlikely to be familiar to other researchers. Taking the same approach and applying it to a range of different toy problems here, with larger models (at least 10’s of units per layer for a two layer or deeper network) would go a long way addressing some of those points. (e.g.: multiple different small regression or classification problems; perhaps even a reduced scale version of MNIST to help connect with something very familiar.) Likewise, it would be interesting to consider a more thorough treatment of gradient based optimization. In particular, to see how KOT might compare to more sophisticated variants of gradient descent.

Correctness: To be best of my knowledge, the claims and methods are correct. However, I had not come across KOT prior to this paper -- and so have low confidence in this assessment.

Clarity: The paper is mostly clear and well written.

Relation to Prior Work: I believe that relevant prior work is cited. However, I had not come across KOT prior to this paper -- so have low confidence in this assessment.

Reproducibility: Yes

Additional Feedback: As noted above, one of the biggest drawbacks of this very interesting work at present is the very limited scope of the demonstrations. I believe this should be easy to address, and were this done I would feel comfortable increasing my score. (I would ideally like to see: (a) implementation on larger networks (this doesn't need to be massive, but 4-hidden units is really tiny); (b) on different types of problem (since the learning dynamics encountered will depend on the loss surface, and this will be problem dependent), and (c) with comparison to different gradient-descent baselines (vanilla (S)GD has many known flaws, but more sophisticated methods perform better and would be the ultimate targets to beat in terms of efficiency). It would also be useful to see more detailed empirical study regarding the choice of the window (t1-t2) used to collect the data to inform the operator approximation.) There are a couple of details that I would like to see to help improve reproducibility. - The authors should also give complete details of the gradient-based baseline in the appendix. - While not required, given the relative simplicity of the approach it could be helpful to provide code/pseudo-code in an appendix to help engage a broader audience. In terms of related work, there are a couple of more tangential directions that come to mind where connections could potentially be made / that may be interesting for the authors to consider. (N.b.: these are merely suggestions that might be useful to consider -- this is not meant to suggest they must be cited.) - Work related to the “Lottery Ticket Hypothesis” -- it seems that there could be interesting connections here: the current KOT work (to my understanding) has a burn in period, then a period in which data is collected; this data is then used to form an approximate operator, which can then be used to perform linear evolution. There may be connections related to initial stages of gradient descent identifying subspaces in which most of the parameter evolution will occur (i.e. containing the lottery ticket weights). (e.g.: Frankle, J., Dziugaite, G.K., Roy, D.M. and Carbin, M., 2019. Linear mode connectivity and the lottery ticket hypothesis. arXiv preprint arXiv:1912.05671.) - Work related to “Synthetic Gradients” -- in which predictive models of gradients are built, based on previously observed gradients. (e.g.: Czarnecki, W.M., Świrszcz, G., Jaderberg, M., Osindero, S., Vinyals, O. and Kavukcuoglu, K., 2017, July. Understanding Synthetic Gradients and Decoupled Neural Interfaces. In International Conference on Machine Learning (pp. 904-912).) - Work related to “Neural Tangent Kernels”. (e.g.: Yang, G., 2019. Wide feedforward or recurrent neural networks of any architecture are gaussian processes. In Advances in Neural Information Processing Systems (pp. 9951-9960).)


Review 4

Summary and Contributions: ==after rebuttal== Thank you for answering my questions, additional experiments, and the pseudocode. Before Rebuttal, I thought that KOT took into account the loss function, but it doesn't seem to use the loss function in Alg.1. The core of the learning seems to be mostly done at t_start without KOT. KOT may perform speeding up by prediction from the past information rather than optimizing NN to find a local optimal solution. If so, presentations including the title may be misleading and the paper needs to be improved greatly. I think it is easy to understand Fig2c by showing the result of KOT in addition to GD (I do not know what the right end of T_eq shows). ==first review== The authors propose a method to apply Koopman operator theory to the training of neural networks. The idea is interesting and novel to this community including dynamical systems, neural networks, and optimization. Although the presentation had a lack of clarity, the method seems to utilize the prediction of the next step with Koopman (or composition) operator rather than the gradient descent (GD). In addition to the above, the contributions of this paper were to empirically show (1) that the Koopman training reduced the number of perceptron training steps required to reach asymptotic performance compared to GD, and (2) Koopman training for feedforward, fully connected, two-layer NN predicted the expected evolution of the weights and biases under GD and achieved a 10x speed-up over the relevant training time compared to GD implementation. However, the experiments only used simple perceptron and two-layer neural networks, and compared to only GD. There is little theoretical suggestion for other advanced network architectures or competitive optimization algorithms. In summary, although the idea and methodology are interesting and related to this community, more clarity of presentation, evaluation, and theoretical suggestion may be required to obtain higher ratings in this conference.

Strengths: The idea of applying Koopman operator theory to the training of neural networks is interesting and novel to this community including dynamical systems, neural networks, and optimization. The claims in this paper are clear. In addition to the above, the contributions of this paper were to empirically show (1) that the Koopman training reduced the number of perceptron training steps required to reach asymptotic performance compared to GD, and (2) Koopman training for feedforward, fully connected, two-layer NN predicted the expected evolution of the weights and biases under GD and achieved a 10x speed-up over the relevant training time compared to GD implementation.

Weaknesses: The presentation had a lack of clarity especially in the core optimization algorithm for Koopman training, which is commented below. The experiments only used simple perceptron and two-layer neural networks, and compared them to only GD. There is little theoretical suggestion for other advanced network architectures or competitive optimization algorithms.

Correctness: The claim that the Koopman operator theory offers the promise of accelerated training, especially in the context of neural networks is partially supported by the experiments, as described above. However, the experiments only used simple perceptron and two-layer neural networks, and compared to only simple GD. There is little theoretical suggestion for other advanced network architectures or competitive optimization algorithms. Moreover, the presentation of the core optimization algorithm for Koopman training was unclear (for the details, see below), thus I cannot evaluate the correctness of the methodology.

Clarity: Although the introduction and related works are clearly written, the proposed method and results are unclear. In Section 3, the authors need to explain the Koopman training algorithm. Estimating from the comparison to GD in the experiment, the method seems to utilize the prediction of the next step with Koopman operator instead of the gradient. Is it correct? This is a critical problem because of the core algorithm. Similarly, I cannot understand Koopman training via decomposition from L133- (about Koopman mode decomposition), which may be redundant (due to future work). More specific unclear points are commented on below.

Relation to Prior Work: The methodological differences from the previous works seem to be clearly written.

Reproducibility: No

Additional Feedback: L94 What does the symbol | of [g_1| ...| g_m] mean? Figure 2 may be confusing. L228 “computed the loss … after T iteration”. However, Figure 2 is inconsistent with the sentence. Is Figure 2 Koopman training loss or GD loss? In both cases, I did not understand T^eq from this figure. I ignored Figure 2. In the perceptron experiment, Koopman training was performed after 100 (GD?) training steps. Does this suggest that Koopman training is better at exploitation than exploration? I want to know the detail in this topic. Furthermore, in the two-layer NN experiment, is this step equivalent to t_2? Where did you specify the value of t_2?

[Author Response · NeurIPS 2020]

Thank you all for your time, feedback, and acknowledging the novelty and potential of our work. The comments
significantly enhanced the manuscript and led us to stronger results. We address our major weaknesses below:
**Clarity (R1-R4)**: We agree that a pseudo-code would have better clarified our method and its key points. The follow-
ing pseudo-code for Node Koopman training has replaced our verbose explanation in Sec. 3. Pseudo-codes describing
other implementations have also been added (Single Weight, Layer, etc. all follow from slight changes to Alg. 1).

---

**Algorithm 1** Node Koopman training

---

Let $\mathbf{w}_j^l(t) \equiv [w_{j1}^l(t) \; w_{j2}^l(t) \; ... \; w_{jN}^l(t) \; b_j^l(t)]^\dagger$ be weights/bias going **into** node $j$ of layer $l$ after $t$ iterations of
training. For each node $(l, j)$, we wish to find an operator $\tilde{U}$ that satisfies $\mathbf{w}_j^l(t + 1) = \tilde{U}\mathbf{w}_j^l(t)$. Choose $t_1 < t_2$, and
$T > 0$. Train the NN of choice for $t_2$ training iterations and record all $\mathbf{w}_j^l(t)$ from $t_1$ to $t_2$. For each node $(l, j)$:

1. Define matrices: $F = \begin{bmatrix} \mathbf{w}_j^l(t_1) & \mathbf{w}_j^l(t_1 + 1) & \cdots & \mathbf{w}_j^l(t_2 - 1) \end{bmatrix}$ & $F' = \begin{bmatrix} \mathbf{w}_j^l(t_1 + 1) & \mathbf{w}_j^l(t_1 + 2) & \cdots & \mathbf{w}_j^l(t_2) \end{bmatrix}$.

2. Compute $\tilde{U} = F^+ F'$, where $F^+ = (F^\dagger F)^{-1} F^\dagger$.

3. Use $\tilde{U}$ iteratively to predict the evolution of $\mathbf{w}_j^l$ from training iteration $t_2$ to training iteration $t_2 + T$.

---

**(R2, R4)** Sec. 2 is broken into subsections for ease of reading and the Koopman mode decomposition is now discussed
in the Conclusion. **(R4)** We correct a typo in Supplement Table S2 - $t_{\text{start}}, t_{\text{stop}}$ should be $t_1, t_2$. We agree that Fig.
2b and 2c overly complicate the discussion. They have been replaced by clearer text explanations, while Fig. 2e
has been better labeled. $T^{\text{eq}}$ is the number of training iterations needed by the standard optimizer to achieve the loss
performance obtained by $T$ iterations of Koopman training (both are with respect to $t_2$). A Koopman training attempt
is considered successful if $T^{\text{eq}}/T > 0$, with values near 1 implying Koopman training accurately approximated the
optimizer in terms of loss performance. Fig. 2e was meant to highlight that our original result of $T^{\text{eq}}/T = 0.99$ was
due to our method's accuracy in predicting the evolution of individual weights/biases in training iteration time. The
x-axis of Fig. 2e shows the true movement of individual weights/biases from $t_2$ to $t_2 + T$, while the y-axis represents
the error in the predictions made via Node Koopman training (Alg. 1). **(R4)** Koopman training *exploits* existing
weight/bias evolution information to predict their future states (or direction, in the case of the perceptron).

**Generalization (R1-R4):** We agree that our original examples are limited, although Alg. 1 shows that Koopman
training is inherently data-driven and generally applicable to other choices of NN optimizer, size, and problem of
interest. Given the novelty of Koopman training and claims of generalizability, we appreciate the need for more
experiments. **(R3)** We do note that the Hamiltonian NN (HNN) is solving regression problems - their novelty is in the
loss function choice. Further, a major issue for physics inspired NNs (indeed many NNs) is the performance in the
latter half of training: our simple methods are already providing useful cost reductions in this regime.

**(R1-R4)** We replicated experiments with different choices of widths/depths, optimizers, and problems of interest -
including a deeper feedforward NN trained on the full MNIST dataset (adapted from the official Pytorch MNIST
example). We leveraged the platform agnostic nature of Koopman training by using MATLAB for implementation,
which results in even larger speed ups, presumably due to faster/more robust matrix operations (as noted in our original
manuscript, line 272). **(R2-R4)** Koopman training generalized well (Table 1). We re-emphasize that our current
approach **does not** make use of parallelization (even though it is better suited, since only matrix calculations are
involved). No fundamental/technical changes in our methods were made for the new experiments in comparison to the
originals, other than those mentioned in Table 1. Complete details have been provided in the Supplement.

Table 1: Generalization results

| Experiment | Optimizer | Architecture | KOT success | $T^{eq}/T$ | Speed up |
|---|---|---|---|---|---|
| HNN | Adam | 1:4:4:2 | 93% | 0.99 | 103x |
| HNN | Adam | 1:10:10:2 | 84% | 0.85 | 58x |
| HNN | Adagrad | 1:10:10:2 | 92% | 1.00 | 62x |
| HNN | Adadelta | 1:10:10:2 | 100% | 0.98 | 64x |
| HNN | Adadelta | 1:50:50:2 | 92% | 1.00 | 17x |
| MNIST | Adadelta | 784:10:10:10:10 | 100% | 1.00 | 27x |
| MNIST | Adadelta | 784:20:20:20:10 | 100% | 2.00 | 37x |

We hope that these additional comments and experiments enhance the clarity and perspective of our work. We apol-
ogize for any concerns we were not able to address in this response due to limitations of space and have ensured that
every raised point has been addressed in the revised manuscript.

[Meta-Review · NeurIPS 2020]

This paper provides a new perspective on neural network training based on Koopman operator theory (KOT). The paper received mixed reviews (top 50% -> marginally above, reject, marginally above -> accept, marginally below). On the positive side, and despite KOT being very old, the new perspective has a lot of potential: since the KOT is linear, if one can find (or approximate) eigenfunctions, one could compute and analyze training dynamics more easily and make optimization more efficient. On the negative side, the paper is a first step, and needs further development and experimental evaluation to demonstrate the value. Some reviewers also expressed the paper lacks clarity. The rebuttal agrees clarity can be improved and provided a cleared description of the algorithm, which should be incorporated. IMHO, the potential for impact of applying old, but powerful tools from dynamical systems to the analysis of the leaning dynamics of deep networks outweighs the weaknesses.